# Medicare Advantage in Soft Tissue Sarcoma May Be Associated with Worse Patient Outcomes

**DOI:** 10.3390/jcm12155122

**Published:** 2023-08-04

**Authors:** Jennifer C. Wang, Kevin C. Liu, Brandon S. Gettleman, Amit S. Piple, Matthew S. Chen, Lawrence R. Menendez, Nathanael D. Heckmann, Alexander B. Christ

**Affiliations:** 1Department of Orthopaedic Surgery, Keck School of Medicine of the University of Southern California, Los Angeles, CA 90033, USAkcliu@usc.edu (K.C.L.); mschen@usc.edu (M.S.C.);; 2Department of Orthopaedic Surgery, University of South Carolina School of Medicine, Columbia, SC 29209, USA; brandon.gettleman63@gmail.com

**Keywords:** soft tissue sarcoma, insurance, medicare advantage, outcome differences

## Abstract

Medicare Advantage healthcare plans may present undue impediments that result in disparities in patient outcomes. This study aims to compare the outcomes of patients who underwent STS resection based on enrollment in either traditional Medicare (TM) or Medicare Advantage (MA) plans. The Premier Healthcare Database was utilized to identify all patients ≥65 years old who underwent surgery for resection of a lower-extremity STS from 2015 to 2021. These patients were then subdivided based on their Medicare enrollment status (i.e., TM or MA). Patient characteristics, hospital factors, and comorbidities were recorded for each cohort. Bivariable analysis was performed to assess the 90-day risk of postoperative complications. Multivariable analysis controlling for patient sex, as well as demographic and hospital factors found to be significantly different between the cohorts, was also performed. From 2015 to 2021, 1858 patients underwent resection of STS. Of these, 595 (32.0%) had MA coverage and 1048 (56.4%) had TM coverage. The only comorbidities with a significant difference between the cohorts were peripheral vascular disease (*p* = 0.027) and hypothyroidism (*p* = 0.022), both with greater frequency in MA patients. After controlling for confounders, MA trended towards having significantly higher odds of pulmonary embolism (adjusted odds ratio (aOR): 1.98, 95% confidence interval (95%-CI): 0.58–6.79), stroke (aOR: 1.14, 95%-CI: 0.20–6.31), surgical site infection (aOR: 1.59, 95%-CI: 0.75–3.37), and 90-day in-hospital death (aOR 1.38, 95%-CI: 0.60–3.19). Overall, statistically significant differences in postoperative outcomes were not achieved in this study. The authors of this study hypothesize that this may be due to study underpowering or the inability to control for other oncologic factors not available in the Premier database. Further research with higher power, such as through multi-institutional collaboration, is warranted to better assess if there truly are no differences in outcomes by Medicare subtype for this patient population.

## 1. Introduction

Historically, health insurance status has been shown to contribute to differences in clinical presentation, treatments received, and overall survival for various malignancies [1]. The Balanced Budget Act of 1997 allowed for the utilization of private insurers to provide Medicare coverage to eligible patients through Medicare Advantage (MA) plans [2]. While these plans have gained popularity amongst those living in the United States over the past two decades [3], reports have suggested patients may receive suboptimal care as a result of associated delayed care.

Several mechanisms contributing to delayed access to care among Medicare Advantage patients have been cited, including frequent denials of legitimate preauthorization and payment requests [4]. Furthermore, the United States Office of Inspector General reported that MA organizations overturned approximately 75% of prior denials, highlighting specific inefficiencies requiring attention and improvement. However, only a few studies have examined the impact of these insurance plans on healthcare metrics for cancer patients. Raoof et al. assessed the effects of MA health plan networks on patient access to high-volume hospitals for cancer surgery. They found that one-third of older Americans with MA coverage lacked access to high-volume hospitals. The most common patients that comprised the study cohort were those in need of stomach, liver, and pancreas surgery [5]. A study by Kim et al. found that patients with MA plans were 6% less likely to use a top-ranked hospital for cancer care when compared to those with traditional Medicare I plans [6]. This disproportionate utilization of high-quality care by MA patients was thought to be secondary to restricted access, especially those with limited out-of-network benefits [6].

No study to date has examined differences in surgical outcomes for patients with TM or MA coverage in the STS patient population. Due to the complexity of the multidisciplinary care required and the potential for increased complications associated with this type of malignancy, insurance coverage is important to consider when assessing outcomes in patients undergoing surgery for STS [7,8]. Given the potential for disparities in care between MA and TM plans, there is an evident need to examine this in the context of orthopaedic oncology patients. Therefore, the primary aims of the study are to (1) compare the characteristics of patients with lower-extremity STS who underwent resection and (2) assess for differences in postoperative complications based on coverage with TM or MA at the time of surgery. The secondary aim was to assess trends in TM and MA coverage amongst this patient population.

## 2. Materials and Methods

All patients ≥65 years old diagnosed with a lower-extremity STS from 1 January 2015 to 31 December 2021 were identified in the Premier Healthcare Database [9]. The PHD is a U.S. hospital-based, service-level, all-payer database capturing approximately 25% of all hospital admissions in the U.S [10]. As administrative procedural coding is not always specific to oncologic procedures, we first identified all individuals with a diagnosis of a lower-extremity STS using International Classification of Diseases, Tenth Revision (ICD-10) diagnosis codes. Subsequently, individuals within this cohort who underwent surgical resection of a lower-extremity STS were identified using ICD-10 Procedural and Current Procedural Terminology (CPT) codes (Appendix A). This study was exempt from institutional review board approval as all information was de-identified in accordance with the Health Insurance Portability and Accountability Act.

### 2.1. Identification of Study Cohorts

All patients who underwent surgical resection of a lower-extremity STS were categorized per their insurance enrollment status at the time of their index surgery using PHD-specific insurance identifiers: TM, MA, or non-Medicare plan. The TM cohort was the baseline comparator against which the MA group was evaluated. Patients with a non-Medicare plan were utilized only to document trends in insurance enrollment status throughout the study period and were otherwise excluded from statistical analysis.

### 2.2. Independent Variables

Patient demographics (i.e., age, sex, race, length of stay (LOS), year of procedure, and total cost), hospital factors (i.e., size, urban/rural setting, teaching status, and region), and the prevalence of patient comorbidities were assessed.

### 2.3. Study Endpoints

Outcomes assessed were the 90-day risk of thromboembolic complications (deep vein thrombosis (DVT) and pulmonary embolism (PE), stroke, and myocardial infarction (MI), bleeding-related events (acute blood loss anemia, hemarthrosis, hemorrhage, and transfusion), infectious complications (surgical site infection and sepsis), wound complications (wound dehiscence, seroma, and hematoma), medical complications (pneumonia, respiratory failure, acute renal failure, and urinary tract infection (UTI)), readmission, and in-hospital death.

### 2.4. Trends in Insurance Coverage

Trends in the utilization of TM, MA, and non-Medicare plans were assessed from 2015 to 2021. The proportion of patients with each type of insurance plan was calculated by dividing the total number of patients in each cohort by the total number of patients who underwent lower-extremity resection for each respective calendar year.

### 2.5. Statistical Analysis

All patient demographics, hospital factors, and rates of comorbidities were evaluated with descriptive statistics. Independent t-tests and chi-squared analyses for continuous and categorical variables, respectively, were performed to assess for differences between cohorts. Bivariable regression analyses were performed to assess the 90-day risk of all endpoints. Multivariable regression analyses were performed to account for potential confounders. These were patient factors, hospital characteristics, and patient comorbidities that approached a significant difference between cohorts (*p* < 0.100). Statistical significance was defined as *p* < 0.050. All statistical analyses were performed using STATA (version 16.1; StataCorp, College Station, TX, USA).

## 3. Results

### 3.1. Differences in Patient Demographics and Hospital Characteristics

From 2015 to 2021, 1858 patients with lower-extremity STS who underwent resection were identified. Of these, 595 (32.0%) had MA coverage and 1048 (56.4%) had TM coverage. The mean age for the MA and TM groups were 75.8 ± 6.9 and 76.1 ± 7.2 years old (*p* = 0.432), and the cohorts were 46.6% and 52.1% male, respectively (*p* = 0.031). There were more Asian (5.0% vs. 2.3%, *p* < 0.001) and Black (11.3% vs. 5.2%, *p* < 0.001) patients comprising the MA versus the TM group, whereas more White patients (84.8% vs. 77.6%, *p* < 0.001) had TM coverage. Patients of both groups had a similar LOS (3.4 ± 6.4 vs. 2.8 ± 6.3 days, *p* = 0.097). MA patients were more likely to be treated in urban hospitals (96.1% vs. 91.8%, *p* = 0.001) than TM patients (Table 1).

### 3.2. Comorbidity Differences

There were 32 comorbidities analyzed between individuals with MA and TM coverage. Only 2 of the 32 comorbidities resulted in a significant difference between the cohorts: peripheral vascular disease (MA: 7.1%, TM: 4.5%, *p* = 0.027) and hypothyroidism (MA: 16.1%, TM: 12.1%, *p* = 0.022) (Table 2).

### 3.3. Differences in Postoperative Complication Rates

In terms of both bivariable and multivariable analysis, statistically significant differences in the risk of 90-day postoperative complications assessed were not observed between the cohorts (Table 3). The magnitude of the odds ratios trended towards a significantly increased risk of surgical site infection (adjusted odds ratio (aOR): 1.59, 95% confidence interval (95%-CI): 0.75–3.37), PE (aOR: 1.98, 95%-CI: 0.58–6.79), stroke (aOR: 1.14, 95%-CI: 0.20–6.31), and in-hospital death (aOR: 1.38, 95%-CI: 0.60–3.19). The range of the 95%-CI (i.e., the difference between the upper and lower limits) for the aOR of each complication ranged between 0.7 and 6.21 (Table 3). Only three complications had the 95%-CI range approximating 0.7 and 0.8: acute renal failure, acute blood loss anemia, and readmission.

### 3.4. Insurance Coverage Trends

In 2015, 31.7% of patients were covered with MA and 60.0% with TM (Figure 1). In 2021, 46.6% of patients had MA insurance, and 42.4% had TM insurance. This represents a relative 47% increase in the proportion of patients with MA and a 29% decrease in those with TM.

## 4. Discussion

This is the first study to analyze the impact of MA coverage on outcomes in Medicare-eligible patients with lower-extremity STS following resection. In this study, 63.8% of patients had TM coverage, while 36.2% had MA coverage. The proportion of patients with MA increased by 15% from 31.7% in 2015 to 46.6% in 2021, while the proportion with TM decreased by 17.6% from 60.0% in 2015 to 42.4% in 2021. While the differences in postoperative outcomes observed between those with TM and MA coverage were not found to be statistically significant, we believe the wide range of confidence intervals observed (i.e., greater than 0.7–0.8) make it likely this was secondary to study underpowering rather than a true lack of difference [11]. The implications of these data include a potential seven-fold greater risk of PE, a six-fold greater risk of stroke, and a three-fold greater risk of in-hospital mortality for MA patients than TM ones—all of which underscore the importance of further study on this topic.

The results of the present study suggest a potentially increased risk of infectious and non-infectious complications, including death, in MA patients compared to TM patients with lower-extremity STS. In examining the data of this present study, it is pertinent to emphasize that the lack of statistical significance is best interpreted by assessing the reported 95%-CIs. Should the range of the 95%-CIs have been more narrow, such as between 0.7 and 0.8 per suggestions by The Cochrane Collaboration, it would then be reasonable to conclude that no difference truly existed [11]. No prior studies have examined differences in outcomes for patients with lower-extremity STS by insurance status. Jain et al. examined patients diagnosed with STS of the head and neck, extremities, thorax, abdomen, and pelvis using the National Cancer Database (NCDB) and reported no difference in overall survival in patients ≥65 years old with Medicare compared to those with commercial insurance [8]. The results of the present study suggest Jain et al. may have observed non-significant differences when examining the Medicare cohort because of their inability to examine Medicare insurance plans with further granularity (i.e., TM versus MA), which has been shown to obfuscate true, significant differences [12]. In contrast, Voss et al. found that having government insurance and being uninsured were associated with an increased risk of death compared to having private insurance, respectively, in patients with both Stage IIA and Stage IIB/III extremity and superficial trunk STS [13]. Additional research with a larger sample size is therefore warranted to better assess the relationship between Medicare plan subtype and outcomes following resection in patients with lower-extremity STS.

Insurance coverage has been a well-documented factor affecting cancer outcomes, but data regarding the STS patient population are heterogeneous. Prior studies examining the impact of insurance status on STS patient outcomes analyzed STS along with primary bone sarcoma [14,15,16,17,18], included pediatric and adolescent populations [15,16,17,19], examined STS without delineation of localization to the trunk versus extremities [8,13,20], or examined insurance coverage broadly [21,22]. To the authors’ knowledge, this is the first study to examine lower-extremity STS outcomes in the context of Medicare plan subtypes. The gap in this literature is highlighted by a 2014 literature review published by the Kaiser Family Foundation evaluating studies comparing TM and MA plans regarding differences in care provided [23]. At the time of the review, only five studies examined the impact of Medicare enrollment on cancer diagnosis and outcomes, and none used data from the current decade [24,25,26,27,28]. The authors of the review concluded that though the health plan type did not demonstrate a differential impact on patient outcomes, the heterogeneity and age of the data available precluded definitive conclusions overall. Of relevance, none of the studies examined orthopedic oncology specifically. The steady increase in MA enrollment since the early 2000s, alongside data from other specialty areas suggesting differential outcomes associated with MA and TM plans, highlights the importance of disaggregating the umbrella “Medicare” category used in clinical research [3,29,30].

The impact of insurance differences on oncologic outcomes in the United States may be influenced by other sociodemographic factors, such as patient race. In this present study, a significantly greater proportion of Black, Asian, and Hispanic patients comprised the MA group than the TM group, which had more White patients. Alamanda et al. examined patients with STS of the pelvis and extremities from 2004 to 2009 and reported that Black patients were more likely to die from their sarcoma than White patients at 5 years (23% vs. 16%, *p* < 0.001) [31]. While multivariable analysis demonstrated that Black race alone was not an independent predictor of death due to disease, they also found that Black patients, compared to White patients, had larger tumor sizes on presentation (10 cm vs. 8 cm, *p* < 0.001) and had lower rates of radiation (47% vs. 41%, *p* = 0.024) and surgery (80% vs. 85%, *p* = 0.002) [31]. Other studies have reported similar incidence-based mortality between White and Black patients [32]. Voss et al. examined the NCDB and found that Hispanic patients with Stage IIB/III extremity and superficial trunk STS were more likely to receive treatment not adherent to guidelines set forth by the National Comprehensive Cancer Network, which was associated with increased mortality risk at 5 years [13]. Taken together, these data suggest the possibility that differential insurance enrollment may exacerbate existing disparities in oncologic outcomes between racial groups—that is, different social determinants of health impact clinical outcomes, likely in independent and interrelated ways. Further research investigating this is warranted and should examine differences in outcomes using the intersection of race and insurance status rather than each in isolation.

This study has several limitations that should be discussed. First, as a retrospective study utilizing a national database, it is reliant on the accuracy of coding for each patient encounter. The size of the data analyzed, however, makes it such that the integrity of the data is likely not affected to a significant degree by systematic error. Second, the stage of disease at presentation was unable to be obtained for analysis. To mitigate the effect of this, only patients who underwent resection were included, effectively excluding those with late-stage, non-resectable disease or those treated in a palliative manner after presenting with widely metastatic disease. This likely has the benefit of excluding sicker patients, which otherwise may introduce uncontrollable confounders to the data. Third, the receipt of neoadjuvant or adjuvant chemotherapy or radiation therapy was also unavailable through the PHD.

There are several strengths to this study that warrant mention. First, this is the first study to examine outcomes in patients with STS of the extremities as a function of insurance status. Studying STS is challenging, as the overall low prevalence makes adequately powering statistical analyses difficult. As such, studies have often utilized heterogeneous STS cohorts comprising diseases of various anatomic sites. This present study represents one of the largest studies to date of patients with STS of the lower extremity in isolation. While this likely contributed to study underpowering, we believe maintaining homogeneity in the study cohort is a strength that provides an impetus for future research into this topic. Furthermore, this is also the only study to date that evaluates patients with STS by comparing subtypes of Medicare coverage, which is becoming an increasingly relevant distinction [29].

## 5. Conclusions

In this first study on the impact of Medicare Advantage coverage on outcomes in Medicare-eligible patients with lower-extremity STS following resection, a majority of patients had TM coverage, while approximately one-third had MA coverage. When assessing for differences in postoperative outcomes, odds ratios trending towards significantly increased risk of infectious and non-infectious complications, including mortality, with MA were observed, although statistical significance was not achieved. These results should be interpreted in the appropriate context: the absence of statistical significance in the context of wide confidence intervals should not be interpreted as a lack of difference but rather as potentially being secondary to study underpowering. Further investigation into the role of the Medicare insurance subtype on orthopedic oncologic outcomes is warranted.

## Figures and Tables

**Figure 1 jcm-12-05122-f001:**
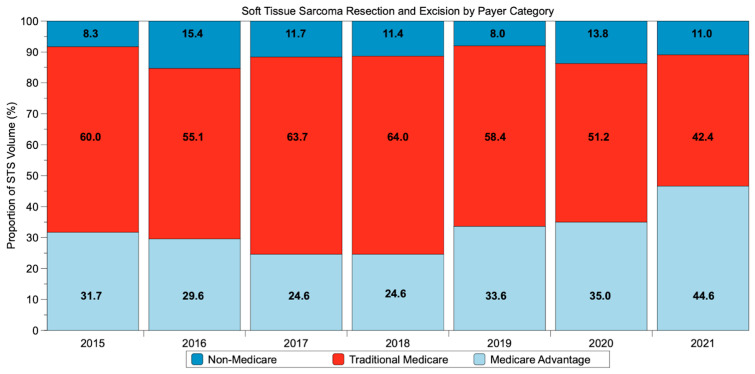
Trend in payor type for patients with lower-extremity soft tissue sarcoma undergoing resection from 2015 to 2021.

**Table 1 jcm-12-05122-t001:** Demographic differences between the patient cohorts.

		Medicare Advantage n = 595		Traditional Medicare n = 1048		*p*-Value
		Average, Standard Deviation		Average, Standard Deviation		
Age (years)		75.8 ± 6.9		76.1 ± 7.2		0.436
Length of Stay (days)		3.4 ± 6.4		2.8 ± 6.3		0.097
Total Cost		13,067.77 ± 18,539.77		11,011.04 ± 15,392.91		0.016
		N	%	N	%	
Male Sex		277	46.6	546	52.1	0.036
Race	Asian	30	5.0	24	2.3	<0.001
	Black	67	11.3	54	5.2	
	Other	20	3.4	70	6.7	
	Unknown	16	2.7	11	1.0	
	Caucasian	462	77.6	889	84.8	
Bed size	<100	18	3.0	33	3.1	0.405
	100–199	33	5.5	90	8.6	
	200–299	49	8.2	92	8.8	
	399–399	65	10.9	103	9.8	
	400–499	83	13.9	145	13.8	
	>500	347	58.3	585	55.8	
Rural vs. Urban	Rural	23	3.9	86	8.2	<0.001
	Urban	572	96.1	962	91.8	
Teaching Status	No	206	34.6	353	33.7	0.837
	Yes	389	65.4	695	66.3	
Region	Midwest	127	21.3	199	19.0	0.747
	Northeast	104	17.5	206	19.7	
	South	241	40.5	440	42.0	
	West	123	20.7	203	19.4	

**Table 2 jcm-12-05122-t002:** Comorbidity differences between TM and MA patients.

	Medicare Advantage N = 595		Traditional Medicare N = 1048		*p*-Value
	N	%	N	%	
Congestive Heart Failure	45	7.6	70	6.7	0.500
Myocardial Infarction	29	4.9	53	5.1	0.870
Valvular Disease	7	1.2	9	0.9	0.529
Pulmonary Circulation Disorders	10	1.7	14	1.3	0.576
Chronic Pulmonary Disease	89	15.0	127	12.1	0.102
Peripheral Vascular Disease	42	7.1	47	4.5	0.027
Hypertension	314	52.8	568	54.2	0.578
Complicated Hypertension	80	13.4	132	12.6	0.621
Cerebrovascular Accident	40	6.7	73	7.0	0.852
Hemiplegia/Paraplegia	5	0.8	4	0.4	0.226
Other Neurological Disorders	25	4.2	36	3.4	0.430
Diabetes, uncomplicated	100	16.8	175	16.7	0.955
Diabetes, Complicated	52	8.7	91	8.7	0.969
Hypothyroidism	96	16.1	127	12.1	0.022
Renal Failure	67	11.3	108	10.3	0.546
Liver Disease	11	1.8	23	2.2	0.636
Chronic Peptic Ulcer Disease	2	0.3	4	0.4	0.883
Blood Loss Anemia	7	1.2	4	0.4	0.058
Deficiency Anemia	20	3.4	23	2.2	0.155
Coagulopathy	15	2.5	22	2.1	0.580
Venous Thromboembolism	41	6.9	56	5.3	0.201
Fluid and Electrolyte Disorders	54	9.1	109	10.4	0.388
Rheumatic Disease	12	2.0	32	3.1	0.211
HIV/AIDS	0	0.0	2	0.2	0.286
Lymphoma	4	0.7	7	0.7	0.992
Solid Tumor	595	100.0	1048	100.0	0.828
Metastatic Cancer	48	8.1	68	6.5	0.230
Obesity	71	11.9	115	11.0	0.555

**Table 3 jcm-12-05122-t003:** Complication differences between TM and MA patients.

Complication	Medicare Advantage N = 595		Traditional Medicare N = 1048		Univariate Regression			Multivariate Regression		
	N	%	N	%	OR	*p*-Value	95%-CI	aOR	*p*-Value	95%-CI
Sepsis	15	2.52	25	2.39	1.06	0.864	0.55–2.02	0.98	0.864	0.55–2.02
Surgical Site Infection	15	2.52	15	1.43	1.78	0.118	0.86–3.67	1.59	0.118	0.86–3.67
Pulmonary Embolism	6	1.01	5	0.48	2.12	0.215	0.65–6.99	1.98	0.215	0.65–6.99
Deep Vein Thrombosis	11	1.85	18	1.72	1.08	0.846	0.51–2.30	1.03	0.846	0.51–2.30
Wound Dehiscence	20	3.36	40	3.82	0.88	0.636	0.51–1.51	0.89	0.636	0.51–1.51
Seroma	2	0.34	4	0.38	0.88	0.883	0.16–4.82	0.85	0.883	0.16–4.82
Stroke	2	0.34	4	0.38	0.88	0.883	0.16–4.82	1.14	0.883	0.16–4.82
Pneumonia	7	1.18	10	0.95	1.24	0.669	0.47–3.26	1.12	0.669	0.47–3.26
Respiratory Failure	8	1.34	23	2.19	0.61	0.228	0.27–1.37	0.6	0.228	0.27–1.37
Myocardial Infarction	1	0.17	6	0.57	0.29	0.255	0.04–2.43	0.35	0.255	0.04–2.43
Acute Renal Failure	23	3.87	53	5.06	0.75	0.27	0.46–1.24	0.71	0.27	0.46–1.24
Urinary Tract Infection	20	3.36	25	2.39	1.42	0.246	0.78–2.59	1.24	0.246	0.78–2.59
Hematoma	6	1.01	11	1.05	0.96	0.937	0.35–2.61	0.92	0.937	0.35–2.61
Hemarthrosis	1	0.17	0	0.00	1	-	-	1	-	-
Acute Blood Loss Anemia	56	9.41	100	9.54	0.98	0.931	0.70–1.39	0.96	0.816	0.67–1.37
Hemorrhage	3	0.50	6	0.57	0.88	0.857	0.22–3.53	0.88	0.855	0.21–3.63
Transfusion	32	5.38	54	5.15	1.05	0.844	0.67–1.64	1.04	0.861	0.66–1.66
90-day Readmission	43	7.23	75	7.16	1.01	0.958	0.68–1.49	0.99	0.943	0.66–1.47
90-day In-Hospital Death	11	1.85	13	1.24	1.5	0.326	0.67–3.37	1.38	0.447	0.60–3.19

## Data Availability

Data from this study are available through purchasing the Premier Healthcare Database. https://www.pinc-ai.com/applied-sciences/solutions/?utm_source=google&utm_medium=paid_search&gad=1&gclid=Cj0KCQjwjryjBhD0ARIsAMLvnF9IaNdQAHflub6rZ-zZjoGe3cDjv0k8matB3eO9p09PzLMrekUnaxsaAvzNEALw_wcB (accessed on 2 November 2022).

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
