# Peer review of "Medicare Advantage in Soft Tissue Sarcoma May Be Associated with Worse Patient Outcomes"

_jcm, 2023, doi:10.3390/jcm12155122_

Round 1
Reviewer 1 Report
Thank you for the opportunity to review this paper. The question of the differences in care quality and health outcomes across Medicare Advantage and Traditional Medicare is an important one, and the attempt by this paper to answer the question for a focused sample of patients who underwent resection for cancer is a good start.
I have a few concerns about the paper which I believe suggest it would benefit from revision. These include:
1. The title and abstract strongly suggest that there are substantial issues with MA in terms of care quality for patients undergoing this specific procedure. However, the results shown in the abstract and the discussion at the end of the paper clearly indicate that the findings were not statistically significant and that the study was likely under-powered to detect significant differences. If this paper is to be published, I would strongly recommend softening the title and editing the abstract to more clearly state this important study limitation. While it's not necessarily feasible to rule out the presence of an effect given these findings, it is also not reasonable to suggest so strongly that MA outcomes are much worse compared to TM given the findings presented in this study.
2. I think more could be done in the abstract and introduction to motivate the reasons behind the differences we might expect to see across MA and TM. The abstract starts with a brief note about the potential for more prior authorization requirements in MA, but the Introduction does not go far enough in exploring why this might lead to differential outcomes - especially the outcomes as explored in the paper. Given the prior authorization angle, I was expecting a key outcome to be whether or not there was a delay between diagnosis and treatment, but this outcome was not assessed at all.
3. Some of the differences seen across MA and TM, especially with regard to receipt of surgery in urban vs rural hospitals, may be due to whether or not Medicare beneficiaries had access to MA plans to begin with in certain areas. While I believe MA is broadly available across much of the U.S. by now, some acknowledgement that TM is likely to cover more beneficiaries in rural areas simply due to a lack of entry by MA plans in those areas might be helpful context for readers.
4. I did not understand why the authors took the extra step of identifying the cancer diagnosis, and instead did not simply identify their sample based on the specific surgery being conducted. Given the explanation in the limitations that diagnosis and staging were not easy to establish in the available data, which is why the authors focused on the actual surgery itself, this step in the analytic process does not appear necessary.
5. Regarding the data, more information regarding its source and its structure, as well as the number of Medicare beneficiaries included, would be extremely helpful. I am not familiar with this dataset and I, along with other readers, would likely benefit from some additional context and orientation.
6. I recommend the authors carefully review the methods descriptions to ensure they are accurate. One example of what I believe to be an error is the description of how the authors calculated the percent of patients enrolled in each Medicare coverage type.
7. The numbers in the text do not match the numbers in Figure 1 (specifically with regard to the percent of enrollees in MA in 2021).
The quality of English language is moderate, and I believe the paper would benefit from copy-editing before final publication.
Reviewer 2 Report
The study comprehensively compares the treatment of lower limb soft tissue sarcoma patients with two types of Medicare. The authors studied de differences in surgical outcomes between those populations.
The tables are big. I understand how important the data are. However, I suggest in Table 1 to suppress at least one line in rural or urban and teaching status "yes" or "no". So, maintain only urban and teaching hospitals.
I wouldn't say I like the style of the conclusion. It looks like a repetition of the discussion. The authors must simplify it.
I believe the authors expected to find considerable differences against Medicare Advanced. However, the excellent study the authors did could not prove it. It is not a demerit to the paper.
The trend is by far different from significant. It would sound better if the authors affirmed that based on that data, the Medicare advance isn't related to a significant increase in postoperative sarcoma surgery complications.
Did the authors run an analysis using only 2021 data, where the proportion of MA patients increased?
Author Response
Reviewer #2:
1. The study comprehensively compares the treatment of lower limb soft tissue sarcoma patients with two types of Medicare. The authors studied differences in surgical outcomes between those populations.
Thank you for taking the time to review our manuscript.
2. The tables are big. I understand how important the data are. However, I suggest in Table 1 to suppress at least one line in rural or urban and teaching status "yes" or "no". So, maintain only urban and teaching hospitals.
Thank you for the suggestion. We’ve adjusted Table 1 as suggested.
I wouldn't say I like the style of the conclusion. It looks like a repetition of the discussion. The authors must simplify it.
Thank you for your comment. We have revised the conclusion and believe it is more simplified than before.
Page 10, Lines 248-253: This is the first study to assess the association of Medicare Advantage on the outcomes in Medicare-eligible patients with lower extremity STS following resection. In this study, statistically significant differences in the risk of postoperative outcomes were not achieved. Whether this is due to study underpowering or a true lack of difference is unable to be determined in this retrospective setting. Further investigation into the role of Medicare insurance subtype on orthopaedic oncologic outcomes are warranted.
3. I believe the authors expected to find considerable differences against Medicare Advanced. However, the excellent study the authors did could not prove it. It is not a demerit to the paper.
Thank you for your comment. We agree that the absence of statistically significant differences does not preclude the study from having value.
4. The trend is by far different from significant. It would sound better if the authors affirmed that based on that data, the Medicare advance isn't related to a significant increase in postoperative sarcoma surgery complications.
We appreciate your feedback. We agree it’s important to represent the data accurately: as it stands, no statistically significant differences were observed between patients with MA and TM. However, we also want to make sure the readership does not misinterpret our data by concluding that a lack of statistical significance in one retrospective study means a true lack of difference between the cohorts. As mentioned in our discussion, the range of the 95% confidence intervals observed may suggest study underpowering, despite the PHD being a nationally-representative database. Together, these suggest future, even larger studies (such as multi-institutional collaborations) be conducted to assess this research question further. We have revised the discussion section in the abstract to provide a more tempered message. We believe the discussion section in the manuscript takes care to not oversell the data or its significance. However, should there be a particular portion of that section that the reviewer believes could benefit from further revision, we would happily comply.
Page 1, Lines 27-32: Overall, statistically significant differences in postoperative outcomes were not achieved in this study. The authors of this study hypothesize this may be due to study underpowering or the inability to control for other oncologic factors not available in the Premier database. Further research with higher power, such as through multi-institutional collaboration, are warranted to better assess if there truly are no differences in outcomes by Medicare subtype for this patient population.
5. Did the authors run an analysis using only 2021 data, where the proportion of MA patients increased?
Thank you for your comment. We did not perform individual analyses by year as our cohort sizes were already relatively small when aggregated across study years. For example, an analysis of 2021 alone would only include 132 MA and 120 TM patients, which would likely preclude robust analyses secondary to insufficient statistical power.
Round 2
Reviewer 1 Report
Thank you for your thoughtful responses to my original review. I appreciate the edits made to address my concerns that the Title and Abstract oversold the potential differences in quality between MA and TM.
I do have a few lingering concerns, namely:
1. The mention of the potential for MA to have worse outcomes due to its use of prior authorization is still only appearing in one sentence in the abstract and it not mentioned anywhere else in the paper. I recommend either deleting the mention entirely and just focusing on the overall issue of insurance differences potentially leading to different quality outcomes, or digging further into what differences exist between MA and TM that may give rise to different outcomes. For what it's worth, I would prefer the latter approach as I think it's helpful to provide some hypotheses as to the mechanisms behind the differences we might end up seeing.
2. (Minor) I appreciate the clarification regarding the need to identify cancer diagnosis as the reason for the STS procedure, and would recommend that this clarification be added explicitly in the paper as well to avoid reader confusion.
3. (Minor) I like the new title but believe the correct word is "affect" not "effect".
I have the same comments as on the previous version. It appears that there have not been substantial edits made to the paper, so my previous comments still apply.
